# Research on Signal Noise Reduction and Leakage Localization in Urban Water Supply Pipelines Based on Northern Goshawk Optimization

**DOI:** 10.3390/s24186091

**Published:** 2024-09-20

**Authors:** Xin Chen, Zhu Jiang, Jiale Li, Zhendong Zhao, Yunyun Cao

**Affiliations:** 1College of Energy and Power Engineering, Xihua University, Chengdu 610039, China; 212022085900012@stu.xhu.edu.cn (X.C.); 212022085900023@stu.xhu.edu.cn (J.L.); 212022085800036@stu.xhu.edu.cn (Z.Z.); 212022080700021@stu.xhu.edu.cn (Y.C.); 2Key Laboratory of Fluid and Power Machinery (Xihua University), Ministry of Education, Chengdu 610039, China

**Keywords:** denoising, leakage localization, negative pressure wave, northern goshawk optimization, urban water supply pipelines, variational modal decomposition, wavelet thresholding

## Abstract

In order to enhance the accuracy and adaptability of urban water supply pipeline leak localization, based on the Northern Goshawk Optimization, a novel joint denoising method is proposed in this paper to reduce noise in negative pressure wave signals caused by leaks. Firstly, the Northern Goshawk Optimization optimizes the decomposition levels and penalty factors of Variational Mode Decomposition, and obtains their optimal combination. Subsequently, the optimized parameters are used to decompose the pressure signals into modal components, and the effective components and noise components are distinguished according to the correlation coefficients. Then, an optimized wavelet thresholding method is applied to the selected effective components for secondary denoising. Finally, the signal components that have been denoised twice are reconstructed with the effective signal components, and the denoised negative pressure wave signals are obtained. Simulation experiments demonstrate that compared to wavelet transforms and Empirical Mode Decomposition, our method achieves the highest signal-to-noise ratio improvement of 12.23 dB and normalized cross correlation of 0.991. It effectively preserves useful leak information in the signal while suppressing noise, laying a solid foundation for improving leak localization accuracy. After several leak simulation tests on the leakage simulation test platform, the test results verify the effectiveness of the proposed method. The minimum relative error of the leakage localization is 0.29%, and an average relative error is 1.64%, achieving accurate leakage localization.

## 1. Introduction

Water is the source of life and is essential for the sustainable development of humans and ecosystems [1]. However, the world is challenged by water scarcity, and many regions suffer from inadequate water supply. Pipeline networks are one of the main carriers of urban and community water supply. Leakage of pipes will lead to waste of water resources and affect the normal operation of water supply and the quality of residents’ lives. In addition, pipeline leakage may also cause problems such as ground collapse and road damage, which can cause damage to urban infrastructure and bring great losses to society and the economy [2]. Leaks in water supply pipelines cause a drop in water pressure, which may lead to the infiltration of pollutants and bacteria around the pipeline, thus contaminating the water quality and posing a hazard to human health [3]. Therefore, timely detection of pipeline leakage and accurate location of leakage have become very important, which has important practical significance for reducing water waste and loss and improving the efficiency and sustainability of water supply.

At present, many scholars have carried out extensive research and proposed a large number of solutions to the problems of pipeline leakage detection and leakage location. Based on the classification of the required conditions, water supply pipe leakage detection and localization technologies are mainly divided into hardware-based methods and software-based methods. The former mainly includes the direct observation method [4], cable detection method, in-pipe detection ball method [5,6], tracer detection method [7,8,9], fiber optic detection method [10], infrared imaging method, and so on [11]. Software-based methods mainly include the mass balance method [12], the negative pressure wave method (NPW) [13], the pressure gradient method, the real-time modeling method [14,15], and the acoustic emission method [16]. Among them, the NPW has been widely used because of its low cost, easy operation, long detection distance, and high sensitivity and accuracy of leakage location, so this paper will focus on the negative pressure wave signal to develop the water supply pipe leakage signal noise reduction and leakage location method research.

Leakage signals are usually affected by factors such as fluid noise, circuit noise, and signal attenuation in the pipe, which can interfere with or mask the real leakage signal, making leakage detection and location very difficult [17]. Therefore, it is necessary to perform noise reduction processing on the acquired leakage signal. Traditional signal noise reduction methods mainly include Fourier transform, wavelet transform, Empirical Modal Decomposition (EMD), etc. These methods are effective in reducing noise interference and improving the signal-to-noise ratio, but they also have some disadvantages. For example, the Fourier transform can only handle periodic signals or finite-length aperiodic signals, and the Fourier transform can only analyze the signal in the frequency domain but not directly analyze signals in the time domain, and the resolution of the Fourier transform on signals is limited by the length of the signal and the sampling frequency [18]. Compared with Fourier transform, wavelet transform has certain advantages in signal noise reduction, but wavelet transform needs to choose the appropriate wavelet basis function. Different wavelet basis functions may have different effects on the signal analysis, so in the practical application of different wavelet basis functions, they need to be compared and selected, which will increase the complexity of the algorithm. In addition, the wavelet transform may have leakage phenomena when dealing with non-smooth signals, which affects the effect of signal noise reduction [19]. Multi-scale wavelet analysis can avoid this drawback by increasing the number of layers of the wavelet transform, but this processing will bring higher computational complexity. EMD is an adaptive signal decomposition technique that automatically decomposes into Intrinsic Mode Functions (IMF) without a priori knowledge [20]. However, it lacks a theoretical foundation and may suffer from modal aliasing for non-smooth signals, which leads to inaccurate decomposition results [21,22], producing pseudo-modalities or loss of true modalities. Variational Modal Decomposition (VMD) is an adaptive signal decomposition method that can estimate multiple modes at the same time, has high decomposition efficiency, is suitable for the study of nonlinear signals, and can avoid the problem of modal aliasing. There are two important parameters in the signal decomposition process of VMD: the penalty factor α and the number of decomposition layers K [23]. The penalty factor α determines the bandwidth of the IMF components. When it is small, the bandwidth of each IMF component is correspondingly large, and some components contain signals from other components. When α is large, the bandwidth of each IMF component is small, which may result in the loss of detailed signals during the decomposition process. The number of decomposition layers K determines the number of IMFs generated by the VMD, each representing a different frequency and amplitude component. When K is small, the number of IMFs obtained from the decomposition is small, which may lead to inaccurate decomposition of complex signals that do not capture all frequency components of the signal. When K is large, the number of IMFs obtained from the decomposition is large, which may over-decompose the signal by decomposing noise or subtle variations into additional IMFs, thus making the decomposition results complex and difficult to interpret. Usually, VMD noise reduction methods are based on empirical or iterative trials to determine the optimal parameter combination of K and α; this process is complex and cumbersome.

Considering that the Northern Goshawk Optimization (NGO) is a heuristic optimization algorithm based on the foraging behavior of the northern goshawk in nature, it mimics the strategy of the goshawk when searching for food and solves the optimization problem by simulating the hunting behavior of the goshawk [24]. The algorithm can quickly converge to the global optimum or near-optimal solutions, and compared with other optimization algorithms, the NGO has fewer parameters, making it relatively simple to use and adapt. In order to improve the adaptability of the VMD, this paper proposes an improved VMD optimization algorithm based on the Northern Goshawk Optimization algorithm, aiming to determine the optimal combination of K and α quickly and accurately. This optimal combination is then used to decompose the signal. For the decomposed IMF components, the conventional practice is to directly reconstruct the high correlation components as denoised signals. Direct reconstruction may reintroduce noise or unwanted components in the high correlation components into the final reconstructed signal, which may lead to distortion or distortion of the signal. This may affect the accuracy of subsequent leakage point localization. In order to improve the denoising effect, this paper adopts the wavelet thresholding method for secondary denoising of the decomposed high correlation components and reconstructs the components after the secondary noise reduction to obtain the denoised NPW. According to the denoised signal, the leakage location method is further studied in this paper. The sudden change in the negative pressure wave shows a series of peaks, among which the sudden change at the leakage singularity is the most severe, expressed as the maximum peak of the upper-level component of the lowest frequency component of the signal (subsequently referred to as the sub-low frequency component). Therefore, in order to extract the peak signal accurately, it is necessary to decompose the signal after noise reduction. Using the above characteristics of the NPW signal, this paper makes modal decomposition of the denoised leakage signa and determines the NPW inflection by calculating the location of the maximum peak in the detail signal; finally, leak location is realized.

The innovation points of this paper are summarized as follows:

(1) The Northern Goshawk Optimization is used to optimize the VMD and wavelet threshold, and the decomposed parameter combination of VMD and wavelet threshold can be obtained adaptively, which improves the operation efficiency, and the noise reduction effect is not affected by repeated trial and error. At the same time, VMD and wavelet threshold denoising are combined to form joint denoising and improve the denoising effect.

(2) Because the features of the leakage information exist in the high-frequency part of the signal, the detail signal is obtained by re-decomposition of the denoised signal, and the leakage singularity is obtained by calculating the maximum peak of the detail signal.

## 2. Methodology

### 2.1. Principle of VMD

VMD is a non-recursive, adaptive signal decomposition method based on Wiener filtering, used to decompose complex non-stationary signals into multiple sub-sequences with different frequency scales [25]. Each IMF of the VMD is equivalent to the frequency-modulated and amplitude-modulated signals described by (1):(1)$$u_{k}\left(t\right)=A_{k}\left(t\right)cos\left(\varphi_{k}\left(t\right)\right)$$
where *k* is the number of IMF, *k* = 1, …, K. $u_{k}\left(t\right)$ is the *k*-th IMF. $A_{k}\left(t\right)$ is the instantaneous amplitude of $u_{k}\left(t\right)$ and $A_{k}\left(t\right)\geq 0$. $\varphi_{k}\left(t\right)$ is the instantaneous phase angle of $u_{k}\left(t\right)$. The instantaneous frequency $\omega_{k}\left(t\right)$ of $u_{k}\left(t\right)$ can be obtained by calculating the derivative of $u_{k}\left(t\right)$, $\omega_{k}\left(t\right)=d\varphi_{k}\left(t\right)/dt$.

The core idea of VMD is to construct and solve a variational problem to obtain the optimal decomposition result of the signal, i.e., by finding k modal functions, such that the total bandwidth of all decomposed IMFs is minimized while their sum is equal to the input signal. The specific steps are as follows:

Step 1: For each IMF component, the corresponding resolved signal is obtained by transforming the one-sided spectrum of each mode by Hilbert:(2)$$\left(\delta \left(t\right)+\frac{j}{\pi t}\right)*u_{k}\left(t\right)$$
where $\delta \left(t\right)$ denotes the unit pulse function and ∗ denotes the convolution operation.

Step 2: The center frequency is adjusted by adding an exponential term to each mode, and then the frequency shift method is used to move the spectrum of each mode signal to the fundamental frequency range:(3)$$\left[\left(\delta \left(t\right)+\frac{j}{\pi t}\right)*u_{k}\left(t\right)\right]*e^{-j\omega_{k}t}$$

Step 3: The gradient square L norm of (3) is calculated to obtain the bandwidth of each mode:(4)$$\left\{\begin{array}{c} \underset{\left\{u_{k}\right\},\left\{\omega_{k}\right\}}{\min}\left\{\sum_{k=1}^{K}{\left\|\partial_{t}\left[\left(\delta \left(t\right)+\frac{j}{\pi t}\right)*u_{k}\left(t\right)\right]*e^{-j\omega_{k}t}\right\|}_{2}^{2}\right\}\\ s.t.\sum_{k=1}^{K}u_{k}\left(t\right)=f \end{array}\right.$$
where $\left\{u_{k}\right\}=\left\{u_{1},\ldots ,u_{K}\right\}$ denotes the K IMF components obtained by the VMD to decompose the signal *f*. $\left\{w_{k}\right\}=\left\{w_{1},\ldots ,w_{K}\right\}$ denotes the center frequency corresponding to each IMF, and $\partial_{t}$ denotes the derivative of the function with respect to time.

Step 4: In order to obtain the optimal solution of (4), a penalty factor and a Lagrange multiplier are introduced to transform it into an unconstrained variational problem:(5)$$\begin{array}{l} \Gamma \left(\left\{u_{k}\right\},\left\{\omega_{k}\right\},\lambda \right)=\alpha \sum_{k=1}^{K}{\left\|\partial_{t}\left[\left(\delta \left(t\right)+\frac{j}{\pi t}\right)*u_{k}\left(t\right)\right]*e^{-j\omega_{k}t}\right\|}_{2}^{2}\\ +{\left\|f\left(t\right)-\sum_{k=1}^{K}u_{k}\left(t\right)\right\|}_{2}^{2}+\langle \lambda \left(t\right),f\left(t\right)-\sum_{k=1}^{K}u_{k}\left(t\right)\rangle \end{array}$$
where α ensures the accuracy of the signal reconstruction and λ ensures the strictness of the constraints.

Using the alternate direction method of multipliers (ADMM), the variables $u_{k}^{n+1}$, $\omega_{k}^{n+1}$, and $\lambda^{n+1}$ are alternately updated to solve the unconstrained problem target and obtain the optimal solution to the variational problem. The formula for updating the variables repeatedly each time is:(6)$${\hat{u}}_{k}^{n+1}\left(\omega \right)=\frac{\hat{f}\left(\omega \right)-\sum_{i\neq k}{\hat{u}}_{i}\left(\omega \right)+\frac{\hat{\lambda }\left(\omega \right)}{2}}{1+2\alpha {\left(\omega -\omega_{k}\right)}^{2}}$$
(7)$${\hat{\omega }}_{k}^{n+1}=\frac{\int_{0}^{\infty }\omega {\left|\hat{u}\left(\omega \right)\right|}^{2}d\omega }{\int_{0}^{\infty }{\left|\hat{u}\left(\omega \right)\right|}^{2}d\omega }$$
(8)$${\hat{\lambda }}^{n+1}\left(\omega \right)={\hat{\lambda }}^{n}\left(\omega \right)+\tau \left[\hat{f}\left(\omega \right)-\sum_{k}{\hat{u}}_{k}^{n+1}\left(\omega \right)\right]$$
where ${\hat{u}}_{k}^{n+1}\left(\omega \right)$ is the residual of $\hat{f}\left(\omega \right)-\sum_{i\neq k}{\hat{u}}_{i}\left(\omega \right)$ after Wiener filtering, $\omega_{k}^{n+1}$ denotes center frequency of power spectrum of current mode function, the Fourier transform of $\lambda \left(\omega \right)$ corresponding to $\lambda \left(t\right)$.

Repeat Step 1 to Step 4 until the condition of (9) is met to stop the iteration.
(9)$$\sum_{k}{\left\|{\hat{u}}_{k}^{n+1}-{\hat{u}}_{k}^{n}\right\|}_{2}^{2}/{\left\|{\hat{u}}_{k}^{n}\right\|}_{2}^{2}<\varepsilon$$
where ε is the given solution accuracy.

According to the VMD principle mentioned above, there are two important parameters in the decomposition process of VMD, that is, the number of decomposition layers K and the penalty factor α. At present, the selection of these values is generally obtained through expert experience or trial and error, which is extremely tedious and difficult to find the best combination, thus affecting the noise reduction effect of the signal. On the other hand, for different signals, this parameter selection method is obviously less adaptive. In order to solve this problem, based on the NGO, an improved VMD optimization algorithm is proposed in this paper, which can quickly and accurately determine the optimal combination of α and K.

### 2.2. Northern Goshawk Optimization

#### 2.2.1. Principle of NGO

NGO is a population-based metaheuristic optimization algorithm proposed by Mohammad in 2022, and this algorithm simulates the hunting process of the northern goshawk:Phase 1: Prey identification (exploration phase)

In the hunting process of the northern goshawk, it first randomly selects a prey and then swiftly attacks it. Because this selection occurs randomly within the search space, it enhances the exploration capability of the NGO algorithm. This phase aims to globally explore the entire search space to identify the optimal regions. The northern goshawk’s prey selection and attacking behavior in this stage can be described using (10) to (12):(10)$$\begin{array}{c} P_{i}=Xm\\ i=1,2,\ldots ,N,m=1,2,\ldots ,i-1,i+1,\ldots ,N \end{array}$$
(11)$$x_{i,j}^{new,P1}=\left\{\begin{array}{cc} x_{i,j}+r\left(p_{i,j}-Ix_{i,j}\right), & \;\;F_{p_{i}}<F_{i}\\ x_{i,j}+r\left(x_{i,j}-P_{i,j}\right), & \;\;F_{p_{i}}<F_{i} \end{array}\right.$$
(12)$$X_{i}=\left\{\begin{array}{cc} X_{i}^{new,P1}, & \;\;F_{i}^{new,P1}<F_{i}\\ X_{i}, & \;\;F_{i}^{new,P1}\geq F_{i} \end{array}\right.$$
where $P_{i}$ represents the position of the *i*-th northern goshawk’s prey. $F_{p_{i}}$ is the objective function value of the position of the ith northern goshawk’s prey. *m* is a random integer in the range of [1, N]. $X_{i}^{new,P1}$ denotes the new position of the *i*-th northern goshawk. $x_{i,j}^{new,P1}$ denotes the new position of the *i*-th northern goshawk in the *j*-th dimension. $F_{i}^{new,P1}$ represents the new objective function value of the ith northern goshawk after the first phase update. *r* is a random number in the range of [0, 1]. *I* is a random integer of 1 or 2.

Phase 2: Chase and escape (development phase)

After the northern goshawk attacks its prey, the prey will attempt to escape. Therefore, in the final stage of chasing the prey, the northern goshawk needs to continue to chase its prey. Because the northern goshawk chase speed is extremely fast, almost in any case can catch up with prey and successfully capture. Simulating this behavior helps enhance the algorithm’s capability for local search within the search space. Assuming the hunting activity approximates an attack position with a radius R, this pursuit phase in the second stage can be described using (13) to (15):(13)$$R=0.02\left(1-\frac{T^{\prime }}{T}\right)$$
(14)$$x_{i,j}^{new,P2}=x_{i,j}+R\left(2r-1\right)x_{i.j}$$
(15)$$X_{i}=\left\{\begin{array}{cc} X_{i}^{new,P2}, & \;\;F_{i}^{new,P2}<F_{i}\\ X_{i}, & \;\;F_{i}^{new,P2}\geq F_{i} \end{array}\right.$$
where $T^{\prime }$ denotes the current iteration number. T denotes the maximum number of iterations. $X_{i}^{new,P2}$ represents the new position of the ith northern goshawk. $x_{i,j}^{new,P2}$ denotes the new position of the *i*-th northern goshawk in the *j*-th dimension. $F_{i}^{new,P2}$ is the objective function value of the *i*-th northern goshawk after the update in the second phase.

This paper employs Minimum Sample Entropy (SampEn) as the objective function for the NGO algorithm. SampEn is a statistical measure used to assess the complexity of time series data [26]. It quantifies the irregularity or randomness of data, indicating its degree of disorder. For signal, lower SampEn values imply higher self-similarity of the sample sequence. Conversely, higher values indicate greater complexity in the sample sequence. When the original signal is decomposed into K IMFs by VMD, if the IMFs contain less noise components, the stronger the correlation with the original signal, the smaller the SampEn; and vice versa. SampEn is defined as (16).
(16)$$\left\{\begin{array}{l} d\left[{\overset{•}{X}}_{t}\left(a\right),{\overset{•}{X}}_{t}\left(\beta \right)\right]=\underset{\rho \in \left(0,\beta -1\right)}{\max}\left|\overset{•}{x}\left(a+\rho \right)-\overset{•}{x}\left(a-\rho \right)\right|\\ B_{a}^{E}\left(d\right)=\frac{1}{L-E+1}\sum_{\beta =1}^{L-E+1}num\left\{s\left[{\overset{•}{X}}_{t}\left(a\right),{\overset{•}{X}}_{t}\left(\beta \right)\right]<d\right\}\\ B^{E}\left(d\right)=\frac{1}{L-E+1}\sum_{i=1}^{L-E+1}B_{a}^{E}\left(d\right)\\ SampEn\left(E,d\right)=\underset{L\to \infty }{\lim}\left\{-\ln\left[\frac{B^{E+1}\left(d\right)}{B^{E}\left(d\right)}\right]\right\} \end{array}\right.$$
where *E* denotes the embedding dimension, *L* denotes the length, *s* denotes the space, and *d* denotes the distance.

#### 2.2.2. VMD Based on NGO

Using the above NGO algorithm, this paper proposes an improved VMD method (NVMD), which adaptively finds the optimal combination of decomposition layers and penalty factors. The specific steps are as follows:

Step 1: Input the original leakage signal, initialize the NGO parameters, set the search range of the optimization parameter K to [2~10] and α to [1500~2500].

Step 2: Calculate SampEn to achieve the target.

Step 3: Refresh and include the goshawk’s location to determine the optimal position for it.

Step 4: Based on the current minimum SampEn, update the objective value and the optimal position.

Step 5: Repeat the above steps until the maximum number of iterations is reached. The optimal solution of decomposition layers K and penalty factor α in VMD is obtained.

The flow chart of NVMD is shown in Figure 1.

According to the decomposition layers and penalty factors obtained in the above steps, VMD is used to decompose the signal. In the decomposed signal, the signal noise is reduced by finding out the effective component and the noise component. Considering that the Pearson correlation coefficient has the advantages of strong interpretability and high sensitivity, this paper chooses the Pearson correlation coefficient to characterize the correlation of the decomposed signal. It is defined as [27]:(17)$$cc=\frac{\sum_{i=1}^{n}\left(Z_{i}-\overset{―}{Z}\right)\left(Y_{i}-\overset{―}{Y}\right)}{{\left\{\sum_{i=1}^{n}{\left(Z_{i}-\overset{―}{Z}\right)}^{2}\sum_{i=1}^{n}{\left(Y_{i}-\overset{―}{Y}\right)}^{2}\right\}}^{1/2}}$$
where *cc* represents the Pearson’s correlation coefficient of the signals *Z* and *Y*. $Z_{i}$ and $Y_{i}$ denote the *i*-th observation of the variables *Z* and *Y*. $\overset{―}{Z}$ and $\overset{―}{Y}$ denote the mean values of the variables *Z* and *Y*. *n* is the length of the signals.

The conventional way of signal denoising is to directly reconstruct the high correlation components as denoised signals, and in this paper, in order to improve the denoising effect, we perform a secondary denoising on the selected effective components. Considering that wavelet thresholding removes noise while retaining important features of the signal as much as possible so that the denoised signal maintains a high quality of information [28], this paper selects wavelet thresholding for secondary denoising of effective components. Factors affecting wavelet threshold denoising mainly include the wavelet basis, number of decomposition layers, and threshold function [29]. For the selection of the wavelet basis and the number of decomposition layers, this paper also adopts the northern eagle algorithm for optimization; the specific steps are as follows:

Step 1: Input the noise component and initialize the parameters. Set the wavelet basis function to db2~db10 and the number of decomposition layers to 2~10.

Step 2: Calculate the energy difference between the denoised signal and the original signal to obtain the fitness.

Step 3: Update and add the location of the goshawk to obtain the best position for the goshawk.

Step 4: Update the target value and optimal position based on the current minimum energy difference.

Step 5: Repeat the above steps until the maximum number of iterations is reached.

Step 6: Output the optimal wavelet basis function and the number of decomposition layers, and perform the wavelet threshold decomposition of the signal to complete the secondary noise reduction.

Finally, the NPW noise reduction signal is obtained using the secondary denoised components.

#### 2.2.3. Evaluation Indicators

To quantitatively analyze the denoising effect, this paper employs Signal-to-Noise Ratio (SNR) [30] given by (18) and Normalized Cross Correlation (NCC) [31,32] given by (19) as metrics to evaluate the denoising performance:(18)$$SNR=10lg\frac{\sum_{n\max}^{n=1}P^{2}\left(n\right)}{\sum_{n\max}^{n=1}{\left(P\left(n\right)-Q\left(n\right)\right)}^{2}}$$
(19)$$NCC=\frac{\sum_{n\max}^{n=1}P\left(n\right)Q\left(n\right)}{\sqrt{\left(\sum_{n\max}^{n=1}\left(P^{2}\left(n\right)\right)\right)\left(\sum_{n\max}^{n=1}\left(Q^{2}\left(n\right)\right)\right)}}$$
where $P\left(n\right)$ denotes the original signal, $Q\left(n\right)$ denotes the noise reduced signal.

### 2.3. Simulation Test Verification

In order to verify the effectiveness of the proposed method, we first constructed an analog signal *s* to carry out the simulation test. The specific parameters of the signal are as follows:$$\begin{array}{l} s_{1}=\sin(3\times \pi \times 0.004\times n);\\ s_{2}=5\times \sin(2\times \pi \times 0.019\times n+0.8);\\ s_{3}=7\times \sin(2\times \pi \times 0.013\times n+1.2);\\ s=s_{1}+s_{2}+s_{3};\\ n=[1:3000] \end{array}$$

Gaussian white noise is added to the constructed analog signal *s* to generate a noise-containing analog signal s_noisy with a SNR of 5 dB, as shown in Figure 2, where the black line is the analog signal *s*, and the green line is the signal after adding random noise.

The joint noise reduction algorithm proposed in this paper is used for noise reduction in noisy analog signal s_noisy. The objective function curve of NVMD is shown in Figure 3. As can be seen from Figure 3, the adaptability value is the smallest at the 6th iteration, which is 0.029176, the corresponding number of decomposition layers K = 10, and the penalty factor α = 2500.

The correlation coefficient of each IMF component and *s* after NVMD are calculated, and the results of the correlation coefficient are shown in Table 1.

From Table 1, it can be seen that the correlation coefficients of IMF9 and IMF10 are greater than 0.3, so they are considered as effective components, while the rest are regarded as noise components. The effective components are then subjected to secondary denoising using NWTD. The objective curve of NWTD is shown in Figure 4, and the objective function reaches its minimum value of 0.018911 at the 5th iteration. The corresponding optimal wavelet basis is db10, with a decomposition level of 4 layers.

According to the optimization results, NWTD is used for secondary denoising of the effective components, and the final denoised signal is obtained by reconstructing the secondary denoised components. In order to verify the noise reduction effect, this paper compares the noise reduction effect of wavelet and EMD with that of NVMD. Among them, for the wavelet method, according to the research results of using the wavelet method to achieve pipeline leakage signal noise reduction in the literature [33], the same wavelet parameter setting is adopted in this paper, that is, the “sym4” wavelet is adopted for four-layer decomposition. EMD and NVMD select modal components with correlation coefficients greater than 0.3 for reconstruction, and the rest of the low-correlation components are rounded off. Figure 5 demonstrates the noise-containing analog signal s_noisy (green part of the figure) and the noise reduction results after processing by the four methods (black curves in the figure).

As can be seen from Figure 5a, the signal after wavelet denoising still contains a large amount of noise, and it is difficult to observe the real curve of signal *s*. Figure 5b shows the signal by the EMD; although the signal noise of the EMD has a very large reduction, the noise-reduced signal has been distorted compared with signal *s*. Figure 5c shows the signal curve obtained by NVMD; it can achieve better restoration compared to the noiseless analog signal *s*. However, it may reintroduce noise or unwanted components from the highly correlated components into the final reconstructed signal. The signal processed by our proposed method is shown in Figure 5d. As can be seen from the results in Figure 5d, the method proposed in this paper can better retain the useful feature information in the signal, and the signal curve after noise reduction has better consistency with the signal *s*.

Meanwhile, in order to qualitatively analyze, the SNR and NCC of the four methods are calculated separately, and the results are shown in Table 2.

From the calculation results in Table 2, compared with the other three methods, the SNR and NCC of the proposed method are the highest. It shows that this method has better noise suppression ability.

## 3. Leak Location Based on Improved VMD

### 3.1. Pipe Leakage Location Principle Based on NPW

The schematic diagram of pipeline leakage location based on NPW is shown in Figure 6.

When a leak occurs in the pipeline, the fluid inside the pipeline will flow out of the leakage port, resulting in a pressure drop inside the pipeline. At this time, the liquid at both ends of the leakage port will be affected by the pressure difference to the leakage port. Due to the relatively small leakage port, the fluid through the leakage port speed will increase dramatically, forming a high-speed flow. This fast-flowing liquid will drive the surrounding liquid to form a negative pressure region along the pipe wall to both sides of the propagation, the formation of NPW.

According to this principle, we can install sensors on the pipe to catch the pressure drop generated when the pipe leaks. The moment of this sudden pressure drop is often referred to as the leakage signal singularity [34]. By calculating the time difference of the singular points collected by the two sensors, the distance between the leakage point and the sensor can be calculated by using (20), and the leakage point can be located.
(20)$$L_{A}=\frac{1}{2}\left(L+v\Delta t\right)$$
where $L_{A}$ is the distance between the leakage point and the sensor A at the first end of the pipe. *L* indicates the distance between the sensors at both ends of the pipeline, m. *v* indicates the negative pressure wave speed, m/s, according to the experience of the sampling wave speed of 1000 m/s. Δt indicates the sensor A and sensor B signal data in the leakage signal singularity of the difference between the time point, s.

Assuming that the leakage singularity captured by Sensor A is $n_{1}$, and the leakage singularity captured by Sensor B is $n_{2}$, with a sampling frequency of f (f = 500 Hz), then the time difference between these two signals can be calculated using (21):(21)$$\Delta t=\frac{n_{1}-n_{2}}{f}$$

### 3.2. Leakage Singularity Extraction

According to the principle of leakage location, the accurate extraction of pressure drop point features is very important to improve the accuracy of leakage point location. Obviously, the detailed information contained in the singularity mainly exists in a section of the signal of leakage pressure drop, so this paper uses the decomposition of the negative pressure wave signal to achieve accurate extraction of pressure drop information.

For example, after applying joint noise reduction followed by VMD with three decomposition layers to the leakage signals collected in the laboratory, the time domain diagram of the obtained decomposition results is shown in Figure 7.

From Figure 7, it can be observed that the frequencies of IMF1~IMF3 are arranged from high to low, and the lowest frequency IMF3 reflects the overall outline of the leakage signal, while the relatively high frequency IMF1~IMF2 reflects the detailed characteristics of the leakage signal. Among them, the second-lowest frequency component IMF2 can well reflect the singularity information of the leakage moment. The sudden change in the negative pressure wave signal appears as a series of fluctuation values with large amplitude, among which the sudden change at the leakage singularity is the most violent, which is manifested as the maximum peak value in IMF2. Therefore, the location of the leakage singularity can be determined by calculating the location of this maximum peak.

Therefore, in this paper, the decomposed sub-low frequency component is used to reconstruct the detail signal, and then the maximum peak calculation of the detail signal is calculated to obtain the leakage singularity. Similarly, in order to adaptively determine the optimal combination of decomposition layers and the penalty factor of the leakage signal, this paper adopts NVMD to perform mode decomposition of the denoised leakage signal, reconstruct the sub-low frequency component, and extract the maximum peak value to obtain the pressure drop singularity.

The noise-reduced leakage signal is processed to extract the leakage singularity with the above method, and the detailed signal and the maximum peak value are obtained as shown in Figure 8.

As can be seen from Figure 8, the method proposed in this paper can effectively extract the singularity of the leakage moment (the position shown by the red arrow in the figure), which lays a good foundation for the subsequent leakage location.

## 4. Test Analyses

### 4.1. Test Equipment

Leak pressure and the size of the leak point can affect noise reduction performance. Excessive leak pressure produces a stronger noise signal, and noise reduction performance may suffer at high leak pressures. Larger leaks also typically produce more noise and vibration, while smaller leaks produce less noise, but there are also high demands on detection and processing to ensure that noise can be effectively identified and reduced even at low noise levels. To verify the effectiveness of the algorithm proposed in this paper, a simulation test platform for water supply pipeline leakage is constructed. To simulate real pipeline leakage, the pipeline in the laboratory is connected to a fire hydrant for water supply, and experiments are conducted under the same leakage area conditions using the water supply pressure of the fire hydrant. The detection performance under different leakage areas will be further studied in the next step. Partial testing equipment is shown in Figure 9.

Figure 9a shows the schematic diagram of the simulated leakage test. The data acquisition card in Figure 9b is responsible for collecting the pressure signals sent by the sensors and transmitting them to the host computer for further data analysis. The host computer has an acquisition interface in which the user can set the basic parameters of the data acquisition card, including sampling frequency, number of data points per channel, trigger mode, acquisition channel, etc. Figure 9c shows a partial experimental pipe section. Figure 9d shows some of the experimental equipment. We simulate a leak by opening the faucet, and the NPW signal generated by the leakage is collected by the pressure sensors installed at the pipeline, and the pressure change in the pipeline can be observed through the pressure gauge.

The leakage simulation test platform built above was used to conduct leakage simulation test, and the collected leakage signals without noise reduction are shown in Figure 10a, and the detailed signal obtained by directly processing the original leakage signal is shown in Figure 10b.

From the localization results in Figure 10b, it can be seen that direct localization processing of the raw leakage signal without denoising cannot achieve accurate leakage moment detection. This is because, during localization, it is necessary to extract leakage features that contain the singular points of the leak; however, the noise interferes with and obscures these features. The original leakage signal in Figure 10a contains significant noise (indicated by the spikes in the figure), which leads to numerous peaks in Figure 10b that are not related to the true leakage singular points. Therefore, the negative pressure wave singularity caused by leakage cannot be extracted by using the undenoised signal and thus cannot be accurately located. Therefore, it is essential to first denoise the original leakage signal to improve signal quality, allowing for effective extraction of leakage features and localization of the leak point.

### 4.2. Joint Noise Reduction for Leakage Signals

The joint denoising algorithm proposed in Section 2.2.2 is adopted to denoise the leakage signal collected in the laboratory, and NVMD is adopted to process the leakage signal (taking the leakage signal collected by sensor A as an example). The objective function curve obtained is shown in Figure 11.

As can be seen from Figure 11, convergence is reached when the population is iterated to the third time, and the objective function value obtained at this time is 0.00039525, and K = 10, α = 2500. According to the calculation results, the leakage signal was decomposed and the correlation coefficient between the IMF component and the NPW signal was calculated. The results are shown in Table 3.

From Table 3, it can be seen that IMF10 is the effective component and the rest are noise components. IMF10 is then denoised using NWTD for secondary denoising. the objective function curve of NWTD is shown in Figure 12.

It can be seen from Figure 12 that it converges at the 7th iteration when the objective function value is 5.9886 × 10^−6^. According to the objective function, the optimal parameter wavelet basis is db3, and the decomposition layer is 6. After the effective component is denoised twice with these parameters, the final denoising signal is obtained by reconstructing the second denoised component. The comparison between the original leakage signal and the joint noise reduction leakage signal is shown in Figure 13.

The leakage signal collected by another sensor is denoised by the same method. The comparison between the original leakage signal and the joint noise reduction leakage signal is shown in Figure 14.

### 4.3. Leak Location

Using the leakage signals after the above joint noise reduction, the leakage singularity method proposed in Section 3.2 is used to extract the NPW singularities. The leakage signal after noise reduction is still taken as an example to perform NVMD, and the objective function curve of NVMD is shown in Figure 15.

As can be seen from Figure 15, convergence is reached when the population is iterated to the third time, and the objective function value obtained at this time is 2.4637 × 10^−4^, K = 10, α = 2405. VMD is carried out with the optimal decomposition parameter, and the sub-low frequency component is reconstructed to obtain the detail signal, and the time corresponding to the maximum peak position of the detail signal is then calculated, that is *t*_1_ = 8.946 s. After the signal collected by sensor A is processed, its detail signal and the maximum peak (the position shown by the red arrow in the figure) are obtained, as shown in Figure 16.

The noise reduction another leakage signal is processed in the same way, and the time corresponding to the maximum peak position is *t*_2_ = 8.952 s, and the detailed signal and the maximum peak (the position shown by the red arrow in the figure) of the original signal collected by sensor B are shown in Figure 17.

The obtained maximum peak position is substituted into (20) and (21) to obtain the leakage position. According to the calculation, the distance between the leakage port and the sensor A is $L_{A}$ = 10.93 m, the actual distance between the leakage port and the sensor A is 10.12 m, the absolute positioning error is 0.81 m and the relative positioning error is 2.91%.

In order to better verify the performance of the method proposed in this paper, leakage simulation experiments are carried out for different leakage locations, and the experimental results are shown in Table 4. *L* is the distance between the two sensors, ρ is the number of groups, $L_{A}$ is the actual distance of the leakage point from the sensor, *t*_1_ is the leakage moment of the leakage signal collected by sensor A, *t*_2_ is the leakage moment of the leakage signal collected by sensor B, Δt denotes the time delay between the leakage moments of sensor A and sensor B, $\Delta t=t_{1}-t_{2}$, $Z_{A}$ is the calculated theoretical distance, Ψ is the absolute error, $\Psi =\left|Z_{A}-L_{A}\right|$, and ζ is the relative error, $\zeta =\left|\Psi \right|/L$.

As can be seen from Table 4, for the three leakage ports of a total of nine sets of experimental data localization results, of which, the maximum relative error is 3.3%, the minimum relative error is 0.29%, and the average relative error is 1.64%. The reasons for the above errors may be the attenuation of negative pressure waves propagated through pipelines, as well as errors caused by circuits during the process of sensor acquisition and transmission to the main computer. Even if the signal is denoised, the noise in the signal cannot be completely removed, which means there may still be some noise in the signal. These noises can affect the clarity and feature extraction of the signal, leading to positioning errors. Overall, the method proposed in this paper to maintain the original signal waveform on the basis of a very good noise reduction effect, according to the details of the maximum peak signal for different leaks, has achieved an effective positioning of the leak.

### 4.4. Comparison of Methods

In order to better verify the accuracy of the proposed method for locating the leak, EMD, wavelet and NVMD methods used in Section 2.3 are used to denoising the leakage signals collected in the laboratory. The leak is then similarly located using the location method described in Section 3.2. The localization results obtained are shown in Table 5, where $Z_{1}$ is the localization result of EMD, and $\zeta_{1}$ is the relative error of EMD, $\zeta_{1}=\left|Z_{1}-L_{A}\right|/L$, $Z_{2}$ is the localization result of NVMD, $\zeta_{2}$ is the relative error of NVMD method, $\zeta_{2}=\left|Z_{2}-L_{A}\right|/L$, $Z_{3}$ is the localization result of wavelet and $\zeta_{3}$ is the relative error of wavelet, $\zeta_{3}=\left|Z_{3}-L_{A}\right|/L$.

From the results of the three localization methods in Table 5, both EMD and wavelet did not achieve the localization of the leakage port. Some groups of the NVMD method also did not achieve the localization, and the minimum relative error of the localization results was obtained to be 4.20%, the maximum localization error was 10.48%, and the average error was 6.66%.

The reasons that cause the localization results of the above three methods are analyzed separately, starting with EMD, and taking this set of leakage signals used for joint noise reduction in Section 4.2 as an example, EMD is used for noise reduction, and the noise reduction results obtained are shown in Figure 18.

As shown in Figure 18, the leakage signal denoised by EMD is similar to the denoising results of the simulated signal. Although a significant amount of noise has been removed and the resulting denoised signal curve is very smooth, the denoised signal deviates from the actual negative pressure wave curve. In particular, the singular points containing leakage feature information, as shown in Figure 18b for the leakage signal collected by sensor B, have been lost after EMD denoising, making it impossible to perform subsequent leak point localization. Experiments have demonstrated that the traditional EMD method results in signal distortion after denoising, making leak point localization unachievable.

The two leakage signals in Section 4.2 are also used for noise reduction using the NVMD method, and the noise reduction signals are localized, and the obtained noise reduction signals and localization results are shown in Figure 19.

As can be seen from Figure 19, the leakage signal collected by sensor B realizes the extraction of leakage moments, while the leakage signal collected by sensor A does not. The reason is that the leakage signal collected by sensor A still has a few noises after the NVMD noise reduction process, and the amplitude fluctuation at the moment of 19.658 s is more drastic than the leakage moment, and the peak value of the detail signal extracted by the re-decomposition is larger than that of the leakage moment at the moment of 19.658 s, so the final result locates the moment of 19.658 s as the leakage moment, which results in the leakage signal collected by sensor A not being extracted to the exact leakage moment, and ultimately cannot realize the localization of the leakage port. signal is not extracted to the exact leakage moment, and finally the leakage port localization cannot be realized. Comparing the noise reduction (Figure 13) and localization results (Figure 16) of the leakage signal collected by Sensor A with the joint noise reduction method proposed in this paper, it can be seen that the proposed method, which involves secondary noise reduction, mitigates these fluctuations and thus does not localize the moment of 19.658 s as the leakage moment. Instead, it successfully identifies the actual leakage moment.

Taking the leakage signal of group 6 in Table 5 as an example, the same original leakage signal is processed for noise reduction and localization using wavelet and the proposed method in this paper, respectively, and the obtained noise reduction results are shown in Figure 20 and the localization results are shown in Figure 21.

The wavelet noise reduction effect is worse than the NVMD method, which is reflected in the noise reduction effect of the leakage singularity part, in addition to the small amplitude fluctuation present in the NVMD method mentioned above. As can be observed from Figure 20, compared with the method proposed in this paper, the signal after wavelet noise reduction still contains a lot of noise, especially in Figure 20a, the leakage singularity part. The noise reduction signal shows a sawtooth shape, not continuous decline; the noise will cover the leakage singularity information we need, resulting in Figure 21a, where the wavelet method cannot extract the leakage moments and also cannot realize the leakage port localization.

In summary, the signal after EMD noise reduction is distorted and cannot achieve localization. Compared with the method proposed in this paper, the NVMD method has a slightly worse noise reduction effect, and there are some fluctuations, and these small fluctuations will affect the extraction of the leakage singularity moments, leading to localization failure. The signal after wavelet noise reduction still contains a large amount of noise, which can lead to localization failure. The method proposed in this paper, on the other hand, can exclude the influence of these noises and small fluctuations while preserving the quality of the leakage signal and realizing the leakage port localization.

## 5. Conclusions

In this paper, a joint noise reduction algorithm is proposed based on the North Goshawk optimization, which can achieve adaptive noise reduction for leakage noise. Simulation results show that compared with EMD, wavelet, and NVMD, the joint noise reduction method proposed in this paper has the highest SNR and NCC, and the SNR after noise reduction is improved by 12.23 dB, and the NCC reaches 0.991. Through noise reduction and localization preprocessing of real leakage signals collected in the laboratory, the EMD and wavelet cannot be realized for localization, and the average relative error for the localization of the proposed method is 1.64%, which is improved compared with the average error of 6.66% of NVMD for localization. The average relative error of localization of the proposed method is 1.64%, which is 5.02% higher than the average error of 6.66% realized by NVMD. The results show that the joint noise reduction algorithm has the best effect in eliminating noise while retaining the leakage features in the signal, which improves the signal quality and increases the localization accuracy.

The research work in this article mainly focuses on signal denoising and localization, which is only a preliminary study on water pipeline monitoring. As research progresses, we will explore more complex pipeline scenarios, operating conditions, and monitoring methods to improve the effectiveness of the algorithm.

## Figures and Tables

**Figure 1 sensors-24-06091-f001:**
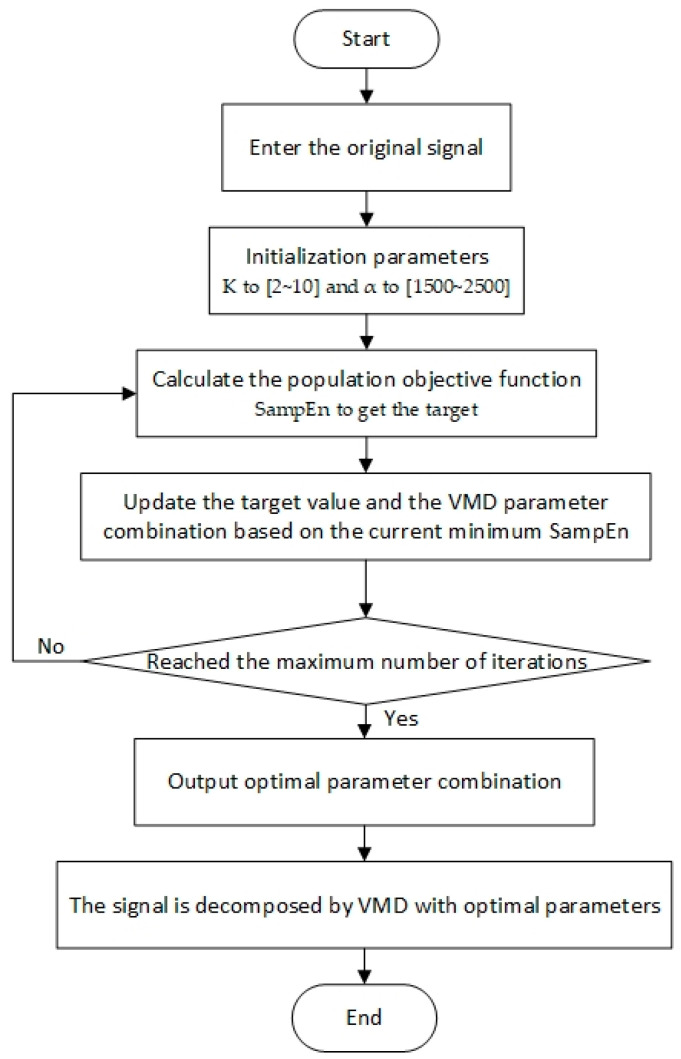
Flow chart of NVMD.

**Figure 2 sensors-24-06091-f002:**
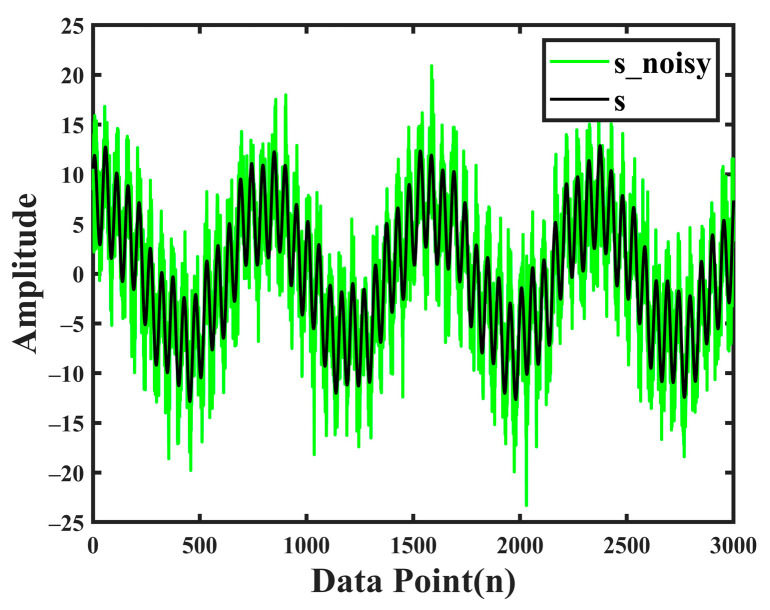
Analog signal *s* and noise-containing analog signal s_noisy.

**Figure 3 sensors-24-06091-f003:**
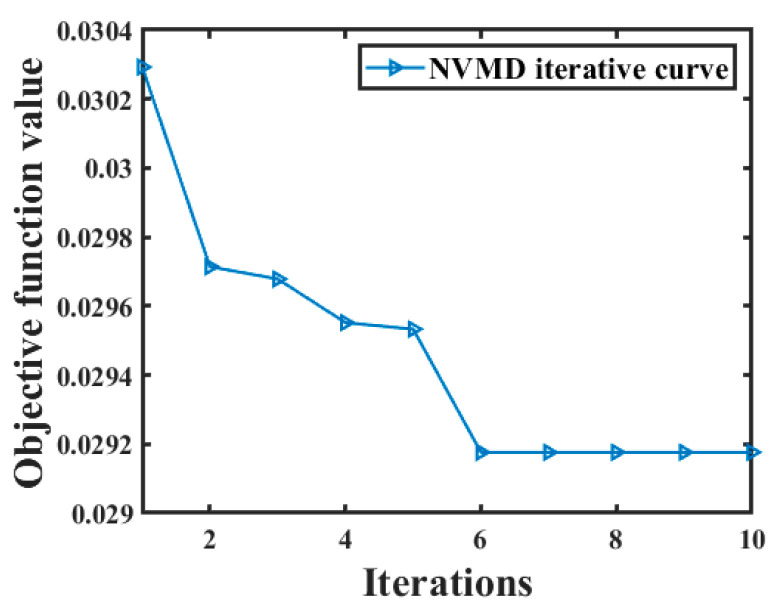
NVMD objective function curve.

**Figure 4 sensors-24-06091-f004:**
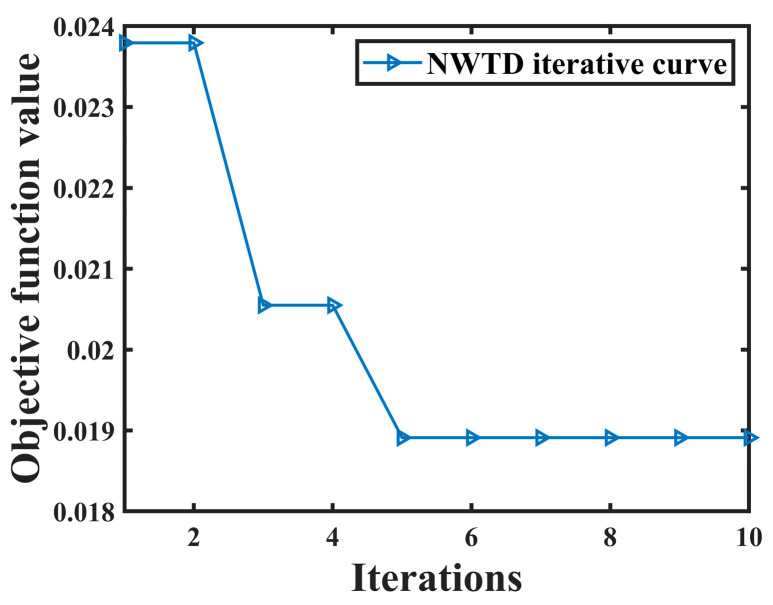
NWTD objective function curve.

**Figure 5 sensors-24-06091-f005:**
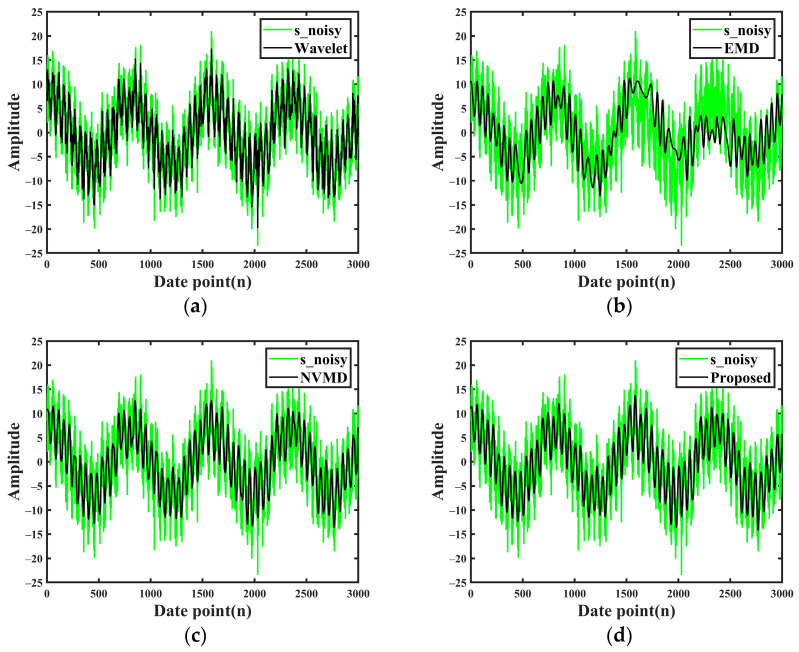
Noise reduction results of the four methods. (**a**) Wavelet noise reduction results; (**b**) EMD noise reduction results; (**c**) NVMD noise reduction results; (**d**) Proposed method noise reduction results.

**Figure 6 sensors-24-06091-f006:**
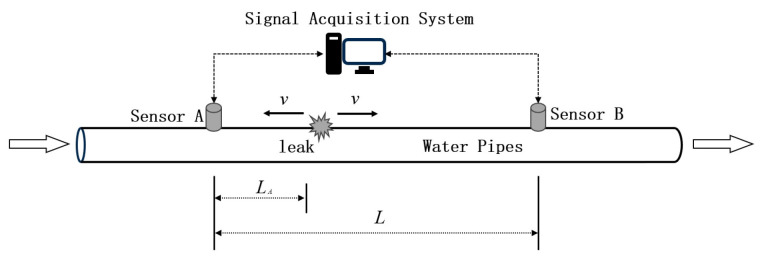
Pipe leakage location schematic (the black arrow indicates that the NPW at the leakage port propagates with wave speed *v*).

**Figure 7 sensors-24-06091-f007:**
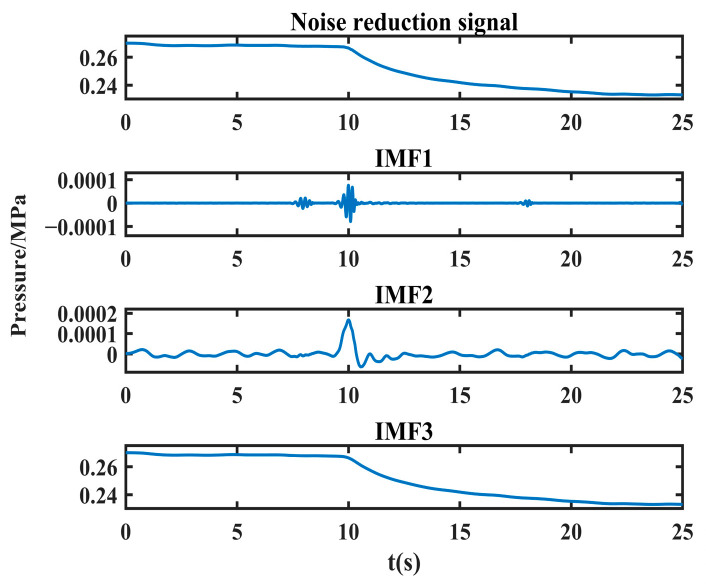
Re-decomposition results of the measured leakage signals after joint noise reduction.

**Figure 8 sensors-24-06091-f008:**
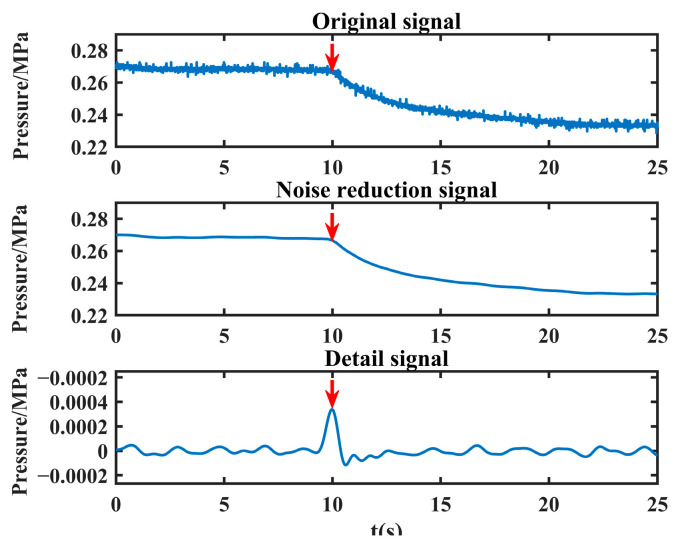
Detailed signal (the red arrow in the figure indicates the maximum peak).

**Figure 9 sensors-24-06091-f009:**
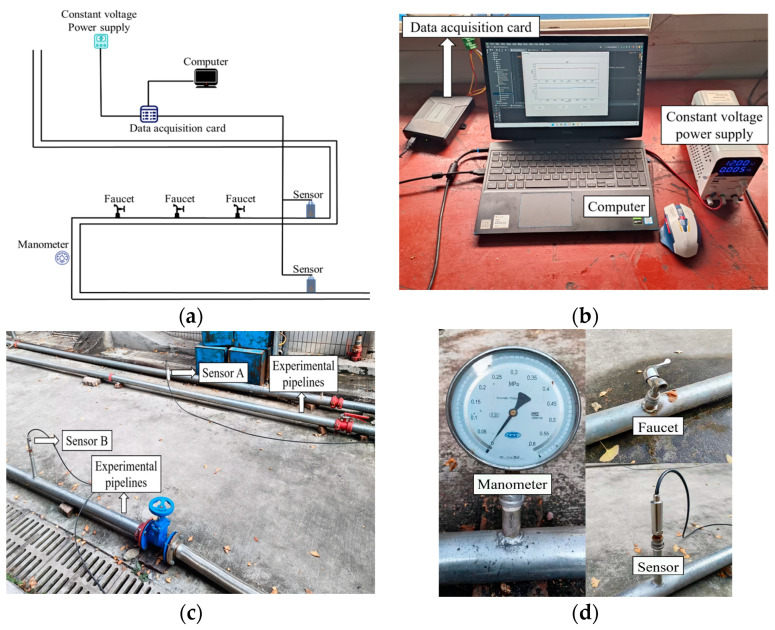
Laboratory equipment. (**a**) Schematic diagram of laboratory equipment; (**b**) Data acquisition card, computer, and constant voltage power supply; (**c**) Partial experimental pipe section; (**d**) Partial experimental equipment.

**Figure 10 sensors-24-06091-f010:**
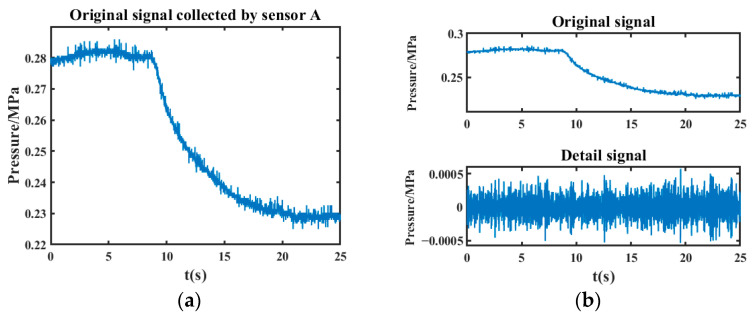
Original leakage signal and its detailed signal. (**a**) Original leakage signal; (**b**) Detail signal.

**Figure 11 sensors-24-06091-f011:**
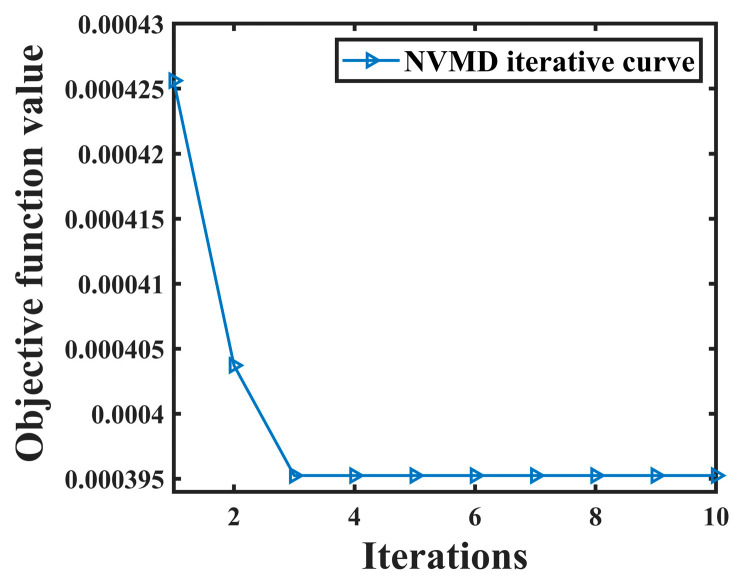
NVMD objective function curve of leakage signal.

**Figure 12 sensors-24-06091-f012:**
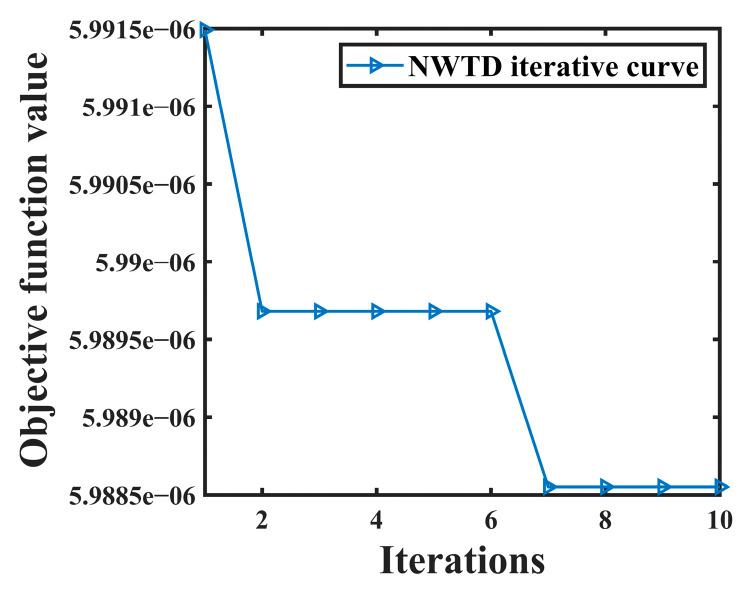
NWTD objective function curve of leakage signal.

**Figure 13 sensors-24-06091-f013:**
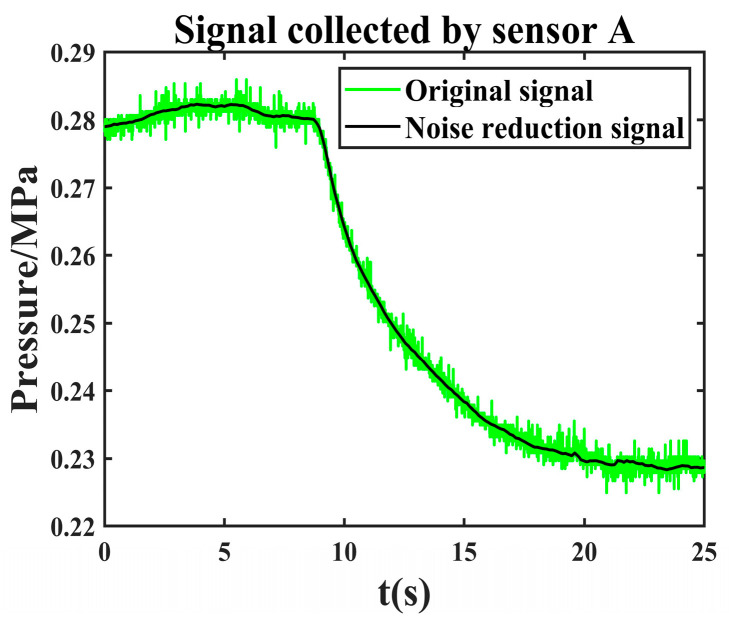
Comparison of original leakage signal collected by sensor A and joint noise reduction leakage signal.

**Figure 14 sensors-24-06091-f014:**
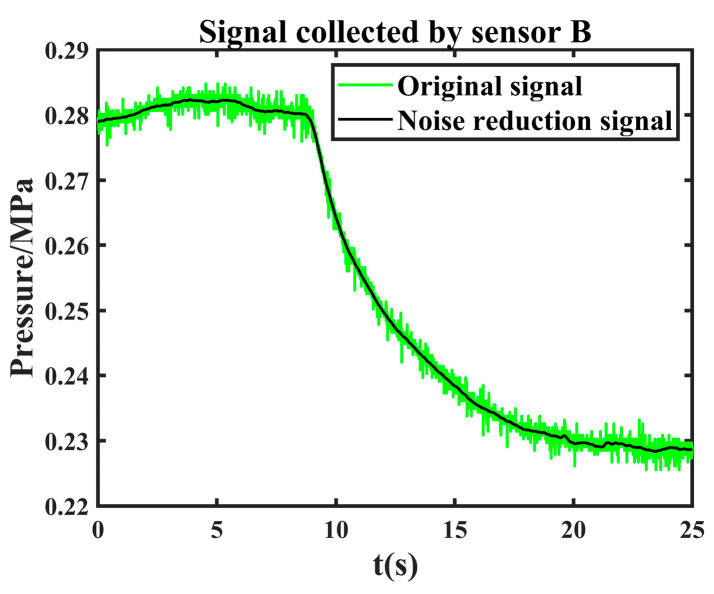
Comparison of original leakage signal collected by sensor B and joint noise reduction leakage signal.

**Figure 15 sensors-24-06091-f015:**
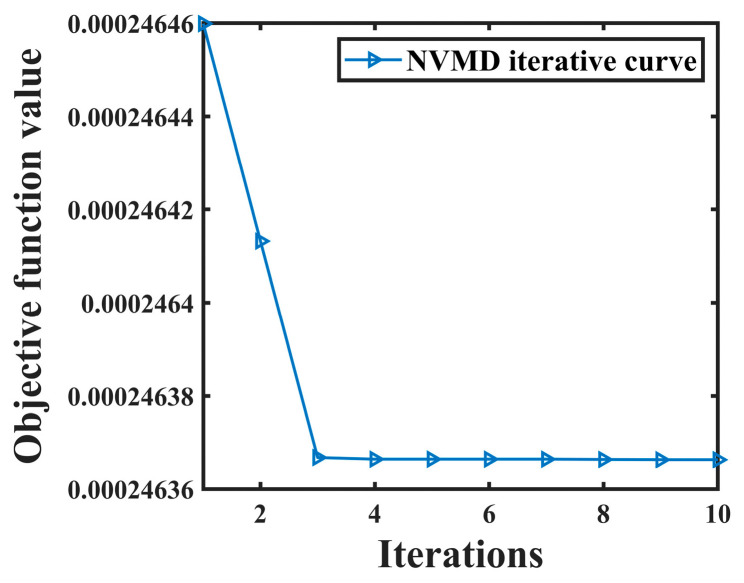
NVMD objective function curves of noise reduction leakage signal.

**Figure 16 sensors-24-06091-f016:**
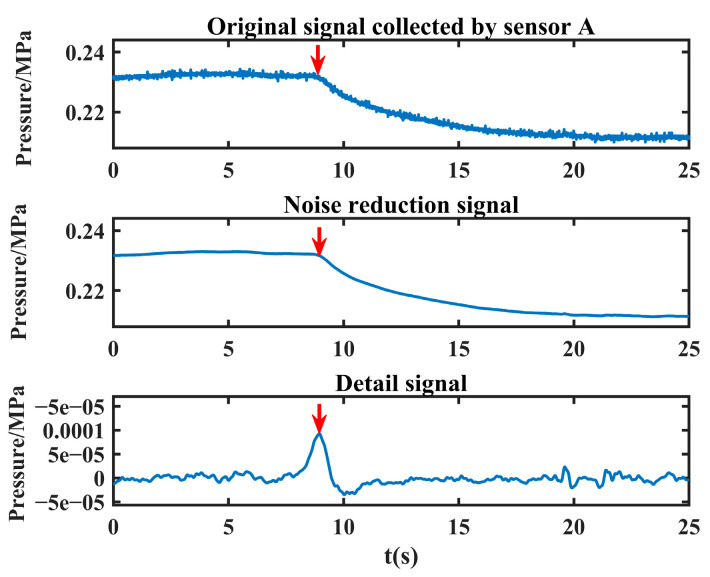
The detailed leakage signal collected by sensor A (the red arrow in the figure indicates the maximum peak).

**Figure 17 sensors-24-06091-f017:**
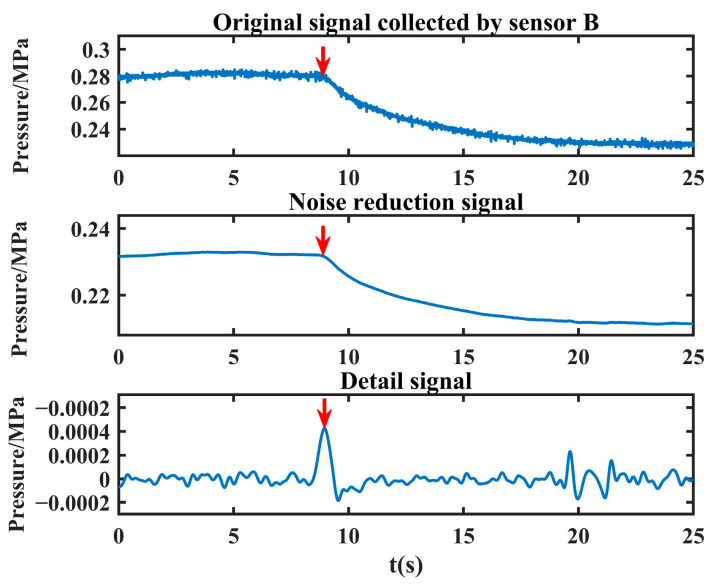
The detailed leakage signal collected by sensor B (the red arrow in the figure indicates the maximum peak).

**Figure 18 sensors-24-06091-f018:**
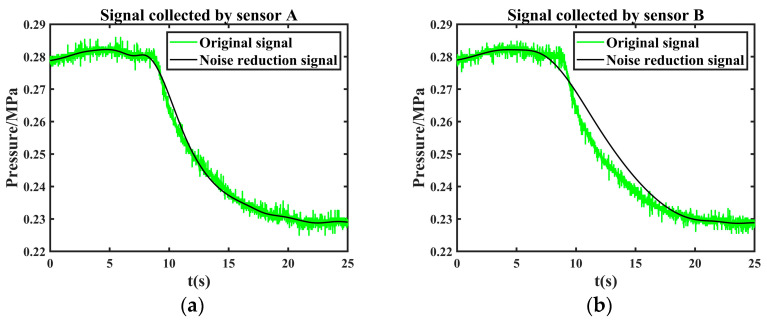
Comparison of raw leakage signal and EMD noise reduction leakage signal. (**a**) Leakage signal noise reduction results collected by sensor A; (**b**) Leakage signal noise reduction results collected by sensor B.

**Figure 19 sensors-24-06091-f019:**
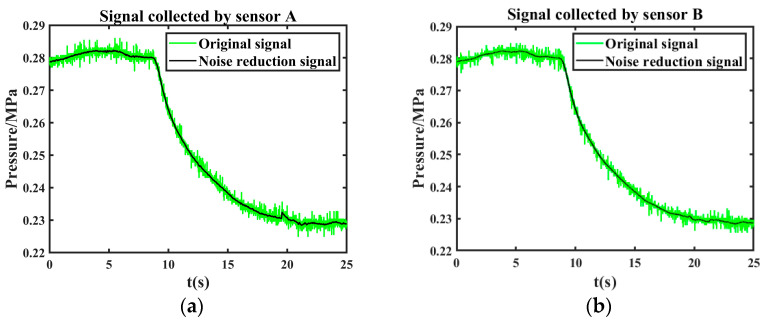
NVMD noise reduction and localization results for two leakage signals (the red arrow in the figure indicates the maximum peak). (**a**) Leakage signal noise reduction results collected by sensor A; (**b**) Leakage signal noise reduction results collected by sensor B; (**c**) Leakage signal localization results collected by sensor A; (**d**) Leakage signal localization results collected by sensor B.

**Figure 20 sensors-24-06091-f020:**
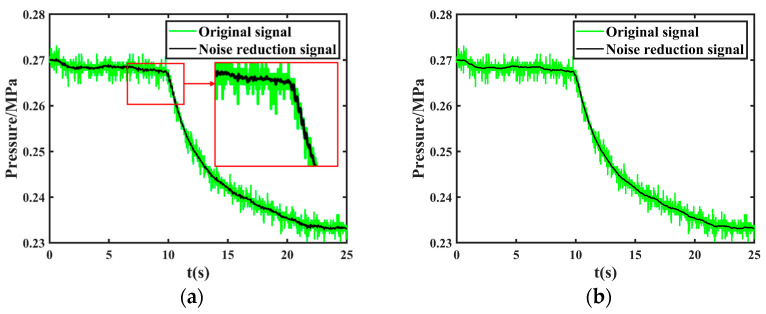
Comparison of noise reduction results for the same leakage signal using wavelet and joint noise reduction. (**a**) Wavelet noise reduction results (the red box on the right is a partially enlarged detail of the red box on the left); (**b**) Joint noise reduction results.

**Figure 21 sensors-24-06091-f021:**
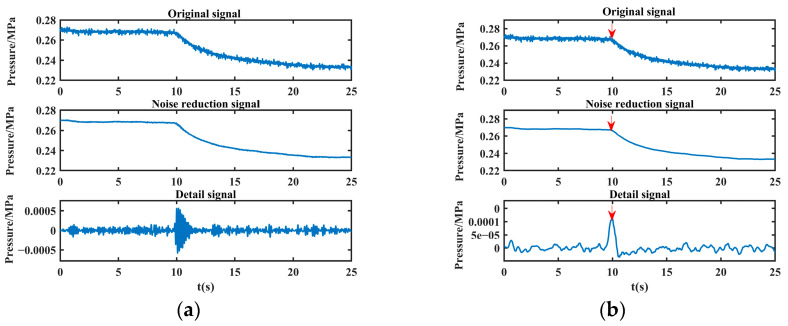
Comparison of localization results for the same leakage signal using wavelet and joint noise reduction (the red arrow in the figure indicates the maximum peak). (**a**) Wavelet localization results; (**b**) Joint noise reduction localization results.

**Table 1 sensors-24-06091-t001:** Correlation coefficient of each IMF component after NVMD.

IMF	*cc*	IMF	*cc*
IMF1	1.2382 × 10^−5^	IMF6	5.9675 × 10^−5^
IMF2	1.3246 × 10^−5^	IMF7	8.2930 × 10^−5^
IMF3	2.0955 × 10^−5^	IMF8	5.3093 × 10^−4^
IMF4	1.0313 × 10^−5^	IMF9	0.5670
IMF5	2.4869 × 10^−5^	IMF10	0.8146

**Table 2 sensors-24-06091-t002:** SNR and NCC obtained by four noise reduction methods.

Noise Reduction Methods	SNR/dB	NCC
Wavelet	11.85	0.968
EMD	4.51	0.810
NVMD	16.09	0.988
Our method	17.23	0.991

**Table 3 sensors-24-06091-t003:** Correlation coefficients of each IMF component after NVMD.

IMF	*cc*	IMF	*cc*
IMF1	−4.25 × 10^−4^	IMF6	9.40 × 10^−4^
IMF2	−5.53 × 10^−5^	IMF7	1.72 × 10^−4^
IMF3	4.72 × 10^−4^	IMF8	8.61 × 10^−4^
IMF4	8.11 × 10^−5^	IMF9	0.0027
IMF5	2.50 × 10^−4^	IMF10	0.9996

**Table 4 sensors-24-06091-t004:** Localization results.

L (m)	Leak	ρ	$L_{A}$ (m)	*t*_1_ (s)	*t*_2_ (s)	Δt (s)	$Z_{A}$ (m)	Ψ (m)	ζ (%)
27.86	1	1	4.01	7.168	7.188	−0.020	3.93	0.08	0.29
		2		9.230	9.248	−0.018	4.93	0.92	3.30
		3		11.480	11.500	−0.020	3.93	0.08	0.29
	2	4	7.1	4.684	4.696	−0.012	7.93	0.83	2.98
		5		8.402	8.414	−0.012	7.93	0.83	2.98
		6		9.982	9.996	−0.014	6.93	0.17	0.61
	3	7	10.12	3.060	3.068	−0.008	9.93	0.19	0.68
		8		3.602	3.610	−0.008	9.93	0.19	0.68
		9		8.946	8.952	−0.006	10.93	0.81	2.91

**Table 5 sensors-24-06091-t005:** Positioning results of different methods.

L (m)	Leak	ρ	$L_{A}$ (m)	*Z_1_* (m)	ζ*_1_* (%)	*Z_2_* (m)	ζ*_2_* (%)	*Z_3_* (m)	ζ*_3_* (%)
27.86	1	1	4.01	—	—	1.93	7.47	—	—
		2		—	—	6.93	10.48	—	—
		3		—	—	5.93	6.89	—	—
	2	4	7.1	—	—	—	—	—	—
		5		—	—	—	—	—	—
		6		—	—	5.93	4.20	—	—
	3	7	10.12	—	—	8.93	4.27	—	—
		8		—	—	—	—	—	—
		9		—	—	—	—	—	—

— means unable to locate.

## Data Availability

Dataset available on request from the authors.

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
