# Peer review of "Research on Signal Noise Reduction and Leakage Localization in Urban Water Supply Pipelines Based on Northern Goshawk Optimization"

_sensors, 2024, doi:10.3390/s24186091_

Round 1
Reviewer 1 Report
Comments and Suggestions for Authors
In the manuscript, authors present a novel joint denoising method to reduce noise in negative pressure wave signals caused by leaks. It is very interesting. The manuscript could be published in the journal after major revision. Below are suggestions that should be addressed by the authors:
1. Whether the signal in Figure 6 is simulated or actually measured?
2. The experiment diagram in Figure 8 does not show the full picture of the experiment, so it is recommended to add a general schematic diagram of the experiment.
3. The main innovation of the paper is noise reduction, so what is the leakage location error calculated by the data without noise reduction, and it is recommended to increase the comparison.
4. The data in Table 5, two of each group are consistent, whether it is a coincidence?
5. What are the factors that generate errors in Table 5 ? Some explanations should be given in the paper.
6. The research on pipeline monitoring in this paper needs to be further strengthened.
Cheng H, Wang F, Huo L, et al. Detection of sand deposition in pipeline using percussion, voice recognition, and support vector machine[J]. Structural Health Monitoring, 2020, 19(6): 2075-2090.
Hou Q, Zhu W. An EKF-based method and experimental study for small leakage detection and location in natural gas pipelines[J]. Applied Sciences, 2019, 9(15): 3193.
Chen B, Hei C, Luo M, et al. Pipeline two-dimensional impact location determination using time of arrival with instant phase (TOAIP) with piezoceramic transducer array[J]. Smart Materials and Structures, 2018, 27(10): 105003.

Reviewer 2 Report
Comments and Suggestions for Authors
Leakage is a great challenge for pipelines, including water pipeline, gas pipeline et al. This paper proposed a novel joint method based on the Northern Goshawk Optimization to reduce noise in negative pressure wave signals caused by leaks. My comments are as follows:
1. Use a flow chart to describe improved VMD method.
2. Will the leakage pressure and leakage point size influence the performance of noise reduction? Please explain.
3. Provide a sketch map for the Laboratory equipment, to help readers get better understanding.
4. Conclusion is so long that it is difficult to get the key points.
5. It is not necessary to introduce the paper structure, but the innovation should be highlighted in the introduction.
6. Could the author make a comparison of the proposed method and available methods? I can get that the proposed method has high accuracy, but how much is the accuracy improved compared with available methods?
Round 2
Reviewer 1 Report
Comments and Suggestions for Authors
The manuscript has been revised and agreed to be published.